# It's all in the music: A systematic review on the effects of musical characteristics on participants' experience and behavior during leisure activities

Céline Daelemans[1,2]*, Casper Bonapart[1], Adriana Leni Smit[1,3]☯, Inge Stegeman[1,3]☯

**1** Department of Otorhinolaryngology and Head & Neck Surgery University Medical Center Utrecht, Utrecht, The Netherlands, **2** Faculté de biologie et médecine, Université de Lausanne, Lausanne, Switzerland, **3** Brain Center, University Medical Center Utrecht Utrecht, The Netherlands

☯ These authors contributed equally to this work.
* celinedaelemans@gmail.com

## Abstract

### Background

Excessively loud music is frequently played at leisure activities, posing significant health risks. However, the lack of consensus on consumers' preferred music settings makes it difficult to implement preventive measures against high noise levels. Therefore, our objective is to systematically evaluate how different musical characteristics influence the experiences and behaviors of individuals engaged in leisure activities.

### Methods

We conducted a search for studies examining the effects of musical characteristics on individuals at leisure activities where the musical experience is of primary focus. The search was performed using the Medline Pubmed, Embase Elsevier, Cochrane, PsychInfo, and ClinicalTrial.gov databases. The exclusion criteria included: leisure activities related to sports, studies evaluating music as a treatment, lab settings, case studies, and participants below 15 years old. The NOS, RoB2, and ROBINS-I tools were used to assess risk of bias. Results relevant to our outcomes of interest were extracted and summarized in tables.

### Results

We identified 2503 studies, of which 37 studies were included for data extraction. The total number of participants in this systematic review was 16843. Among the 37 studies, 23 were observational with the remainder being experimental control trials. Risk of bias in the studies was high. Our findings indicate that musical characteristics such as low frequencies, high groove, high tempo, and live performance enhanced participants' movements and emotions. Excessively high levels, such as those found

**Data availability statement:** All data extraction files are available from the Zenodo database: https://doi.org/10.5281/zenodo.15419911.

**Funding:** Dorhout Mees Stichting provided the funding for this work in the form of salaries for the authors I.S and A.L.S. However, the funder did not play any role in the study design, data collection and analysis, decision to publish, or preparation of the manuscript. The grant number is unavailable. URL to the funder website: https://www.dorhoutmeesstichting.nl/dorhout-mees-stichting/.

**Competing interests:** The authors have declared that no competing interests exist.

in nightclubs, were deemed unnecessary by those exposed. These extreme volumes also caused discomfort and posed a risk to hearing health.

## Interpretation

The high risk of bias makes it difficult to draw conclusions based on the data in this systematic review. Therefore, and in order to inform policy makers, we need adequate randomized controlled trials in order to assess the effects of different levels of loudness on music experience.

**Registration**: PROSPERO registration: CRD42023412634

## Introduction

Over the years amplified music has gained popularity, however, so have dangerous sound levels at musical leisure activities [1]. For instance, studies from the 2000s onwards have reported nightclub sound levels averaging 103.4 dBA, compared to 97 dB in the 1970s [2]. The World Health Organization (WHO) [3] has identified these noise levels as an increasing health threat, as they often exceed the limits set by both the European Agency for Safety and Health at Work (EU-OSHA) and the US-OSHA [3]. Events are not the only sources of loud music that pose a threat as headphones have now become prevalent, especially among adolescents [4].

The consequences of such sound levels pose a threat to hearing health: the WHO estimates that over one billion young people are at risk of hearing loss arising from sound exposure [3]. Noise-induced hearing loss (NIHL) is a type of sensorineural hearing loss that progresses with continual exposure to high decibels [5]. Tinnitus, threshold shifts, and notch hearing loss are widely accepted precursors of hearing damage [4]. Tinnitus can cause a significant personal and societal burden as around 10–20% of affected individuals in the United States report that it severely impacts their quality of life [6].

In response to these health risks, several organizations have attempted to implement prevention policies. In 2022, the WHO published a global standard recommending that venues and events limit their sound volumes to a maximum of 100dB(A) when played continuously for a 15-minute period. This guideline is said to also accommodate for artistic expression and enjoyment of amplified music to be maintained [3]. Nine european countries, such as Switzerland, Belgium, and Germany, have also taken the initiative to write their own detailed regulations [7]. These include: specified sound level limits, real-time sound level monitoring, provision of warning, provision of earplugs, access to quiet zones or rest areas, and restricting access to loudspeakers [7]. Campaigns such as "Know Your Noise", "Dangerous Decibels", and "Don't Lose The Music" have also been set up to educate consumers on loud music and the risks they expose themselves to [2,7]

Despite the regulations already in place, these efforts are limited by the public's preferences [8]. At events like concerts, festivals, or nightclubs the music being played is a key aspect for the clients' attendance [9]. Minor et al. [10] developed a model involving six factors contributing to musical satisfaction. They found that

musical sound was the most influential factor, with participants ranking sound quality and volume as the two most import-ant aspects. Some individuals find that louder music conveys stronger emotions, enhancing the overall musical experi-ence [11]. According to the arousal hypothesis, loud and high-tempo music induces an enhanced behavioral response: "they make me feel happy and energized and I want to turn it up even louder" [11]. Welch and Fremaux [11] mention that emphasized feelings of identity, masking of thoughts, increased intimacy, and easier socializing are other positive out-comes of loud music reported by nightclub attendees. To maximize revenue, venue owners are naturally invested in main-taining customer satisfaction with the music. For example, venue owners could be reluctant to decrease sound levels as it is hypothesized that loud music entices customers to increase their drinking speed, consumption, and attendance [2,10]. Nevertheless, it is unclear whether customers actually want or need such high music levels to have fun. For instance, several studies have demonstrated that individuals prefer slightly lower volumes [2,11,12]

These findings highlight that sound levels are a crucial element of many leisure events, with both positive and negative effects on attendees. The boundary between recommended sound levels for health and audience prefer-ences remains ambiguous. Desired sound levels also vary between individuals and can depend on external factors, such as music preferences, since listeners often choose to play their favorite songs louder than others [13]. However, sound level is not the only musical characteristics that can influence the attendees' satisfaction or experience. Music is composed of various structural components such as frequency, tempo, time stretch, groove and the song's predict-ability [14–17]. There also exist different music genres and ways in which a musical piece can be played or mixed. For instance, certain styles are considered to be groovier, increasing the desire to move [18]. Music has the ability to con-vey powerful messages and emotions to its listeners, often through its lyrics or modality which carries emotional con-notation [16,19]. Additionally, Minor et al. [10] emphasized that listeners also evaluate musician-related aspects such as creativity and interpretation when assessing satisfaction. Given the complexity of these influences, it is challenging to determine how musical elements could be adjusted to meet public health goals without compromising enjoyment. Therefore, in order to gain a comprehensive understanding of how music affects attendees at leisure events, we pro-pose conducting a broad literature review that examines a wide range of musical features and their impact on diverse outcomes.

Understanding the influence that musical characteristics can have on one's experience at events can not only deepen our knowledge of musical satisfaction but also aid policymakers in navigating around the public's opinion to create guidelines for a safer yet equally entertaining experience. For instance, frequencies below 50 Hz have been shown to reduce listeners' preferred sound pressure level [20]. Therefore, adjusting certain musical elements, such as lowering a song's frequency, could help compensate for limitations on sound levels, potentially reducing health risks in nightlife settings while maintaining customer satisfaction. In line with this, our review seeks to answer the following research question: among participants of leisure activities, how do musical characteristics influence their experience and behavior according to the current literature?

## Methods

We will follow the Preferred Reporting Items for Systematic Reviews and Meta-Analyses (PRISMA) 2020 statement as seen in S1 Checklist [21]. The review's pre-registered protocol can be found on PROSPERO International Prospective Register of Systematic Reviews (CRD42023412634).

### Eligibility criteria

Published studies reporting the effects of musical characteristics on individuals attending leisure activities where the musical experience is one of the main aims were considered eligible for inclusion. In this review, musical characteristics were grouped into several categories: sound levels, structural elements of the music (e.g., frequency, time stretch, tempo), music genre, lyrics, emotional connotations of the music, and elements related to the performance or musician (e.g., live, improvised). As our research question focuses on the experience and behavior during the investigated leisure activities,

we considered the ambiance conveyed by the music and the musical performance to be relevant for inclusion. However, any characteristics related solely to the performance and not to the music itself were excluded—for example, lighting, the musician's clothing style, etc. Concerning leisure activities, we only included those where one of the primary reasons for attendance was the music being played, such as nightclubs, festivals, and concerts. For instance, articles focusing on activities such as football matches and the use of headphones during sports were excluded. Furthermore, studies that evaluated the use of music as a treatment were considered to have a non-conforming study design and were therefore excluded. We also excluded studies based on study design if they reversed our intended independent variables and outcomes (e.g., studies that investigated how drug consumption alters music preferences), or if they investigated only a single musical characteristic without including a control group, thereby limiting the interpretability of their findings. Experiments conducted in laboratory settings were only eligible if they aimed to closely simulate real-world leisure activities. To ensure ecological validity, these studies had to replicate key aspects of the experience they sought to model. For instance, an experiment investigating concert attendance needed to be held in an actual concert hall or a comparable venue, and include features such as live music, realistic acoustic conditions, and an audience of engaged participants to reflect the atmosphere of a typical concert environment. Only studies presenting original data, whether qualitative or quantitative, were included. Case studies were excluded. The review focuses on participants aged 15 years or older, hence any studies where more than 50% of participants were below 15 were excluded. Only the participants' experience and behavior were considered outcomes of interest, therefore any physiological outcomes were excluded. Studies focusing on outcomes of hearing health were only included if they also provided additional outcomes that met our inclusion criteria. The different outcomes were used to group studies for synthesis and presentation of results.

### Search strategy and information sources

Medline Pubmed, Embase Elsevier, Cochrane, and PsychInfo were searched on 19 October 2023. To ensure the review reflects the most up-to-date research, the search was repeated on 24 October 2024. The search strategies for each database are available in S1 File. Clinical trials.gov was also searched on both dates for ongoing studies. No filters or limits were used at the time of the search.

### Study selection

The extracted studies from each database were exported to Rayyan [22] and screened independently by two reviewers (CD and CB) for eligibility based on their title/abstract. Then, the full texts of the resulting studies were screened by the same reviewers according to the in/and exclusion criteria.

### Data collection

The data was collected by two reviewers (CD and CB) using a form developed beforehand on Systematic Review Data Repository (SRDR) [23]. Only interventions and outcomes relevant to the research question were extracted. If data was unclear or missing, the corresponding authors of the studies were contacted by email with no reminder being sent in case of no answer, unless contact was previously established. In cases where data remained missing after reaching out to the authors, we extracted any relevant in-text results in the form of quoted text. In this review, three authors were contacted to obtain clarification on unclear data. All of our data was published on Zenodo (https://doi.org/10.5281/zenodo.15419911) [24].

### Data items

The extraction form was composed of eight sections. The design details section included recruitment and sampling procedures, enrolment and start dates, methods used to address missing data, source of funding, and conflict of interest of the study. The experimental groups and their details were recorded in two separate sections: arms and arm details respectively. Group details included participant recruitment and their baseline characteristics (sex, age, and socio-economic

 

status). The details regarding the intervention were extracted into the sample characteristics section. The outcomes relevant to this review and their details (specific measurements and methods of aggregation) were recorded and classified as continuous or categorical in the outcomes and outcome details sections. The total scores of validated questionnaires used to provide a measure of experience or behavior were only extracted if a majority of the questions were relevant to the outcomes of interest. The outcome results for each arm were recorded in a separate section named results. Finally, the last section was dedicated to the risk of bias assessment. Concerning observational studies, the assessment could be completed directly on the SRDR platform. In the case of an experimental controlled trial, the reviewers completed a pdf form of the corresponding risk of bias assessment tool.

### Risk of bias assessment

Two reviewers (CD and CB) worked independently to assess each study. A risk of bias assessment was performed using the Newcastle-Ottawa Quality Assessment scale (NOS) for observational studies, including case-control and cohort studies [25]. The confounders determined as the most important to control for the comparability assessment were: age, hearing health, and frequency of attendance to the researched leisure activity. If all three were controlled a point for comparability to analysis was given. These three factors were selected as comparability criteria because they can vary substantially across studies and may influence participants' preferred sound levels and related behaviors [2]. As no official NOS version exists for cross-sectional studies, an adapted version published by the University of Gent [26] was used. Concerning experimental controlled trials, the risk of bias assessment was performed using the RoB 2/RoB 2 for cross-over trials [27], and the ROBINS-I [28] tools for randomized and non-randomized trials respectively. Only minor discrepancies were encountered in our risk of bias analysis. These were resolved through discussion and consensus between the reviewers. All risk of bias assessment criteria can be found in S2 File.

### Effect measures

When possible, the means, standard deviation (SD), 95% confidence interval (CI), p-value, or percentages and ranges of outcomes were retrieved. If these effect measures were not reported in the article according to the interventions of interest, the corresponding authors were contacted to obtain the raw data. In this review, we used the raw data of four included articles with the approval of the corresponding authors. No second attempts were necessary to get in contact with them.

### Statistical methods

Descriptive analysis was performed on reported data on participants' experience and behavior. Studies that used the same sample population to test the control and experimental groups of their independent variables were eligible for direct comparison between groups. The raw data was obtained and the mean difference between groups (MD) for each participant's outcome results was calculated. Then the mean of all those MDs was calculated for that sample population. Using those results a paired sample t-test to the value of 0 (representing no change) was performed. If the raw data could not be obtained, or the study did not use the same sample population, then a one-sample t-test was performed to assess its difference from the null hypotheses. Significance was determined as a p-value of less than 0.05. Excel and SPSS were used as calculation programs. The results of individual studies and their syntheses are displayed in our results table or summarized in the main text.

## Results

### Study selection

A total of 2503 articles were extracted from the four databases, this number was reduced to 2218 after the removal of duplicates. 84 studies were included after the title/abstract screening. After the full texts of the articles were retrieved and assessed for eligibility, 44 were excluded mainly due to unrelated outcomes or designs that did not align with our inclusion

criteria, and others for alternative reasons (see Fig 1). This resulted in 37 studies included for data extraction. After a brief search on the ClinicalTrial.gov register, no clinical trials corresponding to our eligibility criteria were found.

## Study characteristics

The characteristics of the 37 studies included can be seen in Table 1. Out of the 37 studies, 23 were observational with 2 of them following a cohort design (either retrospective or prospective), 2 case-control retrospective studies, 18 cross-sectional studies, and 1 longitudinal retrospective study. Although Degeest et al. [29] originally conducted a longitudinal study, their primary aim was to perform test-retest evaluations of two questionnaires at different time points. Therefore, only the initial dataset was used to avoid repetition, and the study will be treated as cross-sectional from this point forward. The sample sizes ranged from 16 [11] to 3256 [30], totaling to 16843 participants in this systematic review. In all the studies' population samples, more than 50% of the participants were above 15 except for Theorell et al. [31] where one of the population subgroups was excluded since the sample consisted of children.

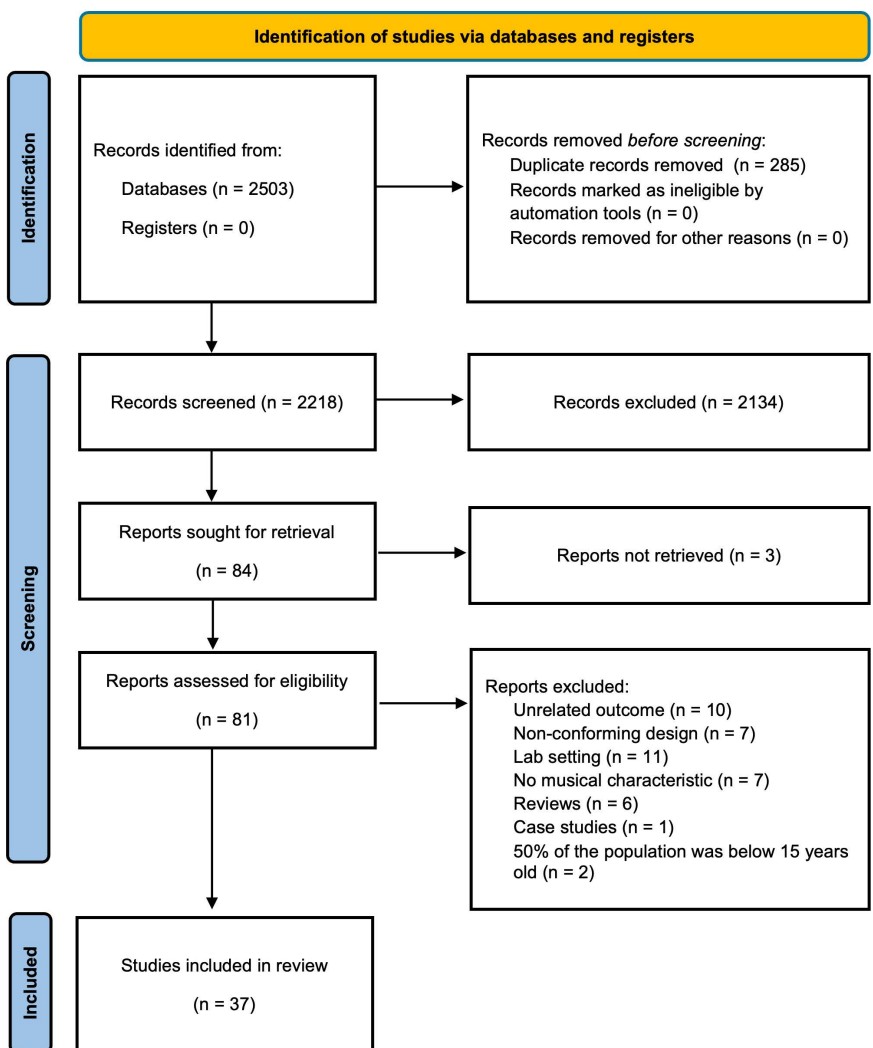

**Fig 1.  Flow chart of article selection process.**

**Table 1. Study characteristics.**

| First author (& second author), year | Country | Study design | N | Age, Mean (SD) | N female (%) | Musical characteristic assessed | Outcome assessed | Outcome measuring tool | Leisure activity |
|---|---|---|---|---|---|---|---|---|---|
| Beach et al. 2019 [2] | Australia | Cohort retrospective | 933 | 30.1 (18.1) | (42.6%) | Sound levels | Effects on hearing health, attitude to loud music, attitude towards HPDs | Large citizen survey | Nightclubs and live music venues |
| Burger et al. 2018 [14] | Germany | Experimental control randomized crossover trial | 30 | 28.2 (4.4) | 15 (50%) | Frequency flux, time stretch, tempo | Movement | Eight camera optical motion capture system | Nightclub (simulated in a lab) |
| Cameron et al. 2022 [32] | Canada | Experimental control non-randomized trial | 62 | 36.4 (12.1) | 32 (51.6%) | Frequency | Movement, enjoyment, body feel, attitude to loud music | Motion capture headband, questionnaires | Live concert (simulated in a lab) |
| Carter & Black 2017 [35] | Australia | Case control retrospective | 210 | 21.8 | 134 (63.8%) | Sound levels | Attitude to loud music, attitude towards HPDs | Questionnaire | General leisure activity with amplified music |
| Coutinho & Scherer, 2017 [36] | Switzerland | Experimental control non-randomized trial | 78 | 33.75 | 58 (74.4%) | Performance type (live vs recorded) | Emotional experiences | Questionnaire | Concert |
| Degeest et al. 2021 [33] | Belgium | Cross-sectional | 242 | 17.5 (1.36) | 164 (66.4%) | Sound levels | Attitude to loud music, effects on hearing health, attitude towards HPDs, | YANS factor 1, questionnaire, BAHPHL factor 5 | General leisure activity with amplified music |
| Degeest et al. 2018 [29] | Belgium | Longitudinal retrospective | 43 | 24.56 (2.98) | 30 (70.0%) | Sound levels | Attitude to loud music, attitude to HPDs | YANS factor 1, questionnaire, BAHPHL factor 5 | General leisure activity with amplified music |
| Dolan et al. 2018 [37] | Canada | Experimental control randomized crossover trial | 22 | -- | 12 (54.5%) | Performance style (improvisatory vs prepared) | Emotional intensity | Verbal questionnaire | Live concert (simulated in a lab) |
| Dotov et al. 2021 [15] | Canada | Experimental control randomized crossover trial | 33 | 26 (7.5)* | 20 (60.6%) | Groove and tempo | Movement, groove, emotional valence, emotional intensity | Motion capture cap, questionnaire | Pre-recorded concert |
| Egermann et al. 2013 [17] | Canada | Experimental control non-randomized trial | 50 | 23 (6) | 21 (42%) | Predictability of the music | Pleasantness, arousal | iPod rating | Live concert (simulated in a lab) |
| Eichwald et al. 2023 [30] | United States | Cross-sectional | 3526 | -- † | 1796 (50.9%) | Sound levels | Attitude to loud music, attitude towards HPDs | Questionnaire | General leisure activity with amplified music |
| Engels et al. 2012 [38] | Netherlands | Experimental control randomized trials | 249 | 21.2 (2.47) | 113 (45.4%) | Music genre | Alcohol consumption | Observation by the experimenters | Bars (simulated in a lab) |
| Engels et al. 2011 [39] | Netherlands | Experimental control randomized trials | -- | -- | -- | Lyrics with alcohol reference | Alcohol consumption | -- | Bars |
| Forsyth, 2009 [40] | Scotland | Cross-sectional | 1847 | -- † | 12025 (55.5%) | Music genre | Alcohol consumption, drug consumption, sexual tension, aggression incidents | Observation by the experimenters | Nightclub |

*(Continued)*

 

**Table 1.** (Continued)

| First author (& second author), year | Country | Study design | N | Age, Mean (SD) | N female (%) | Musical characteristic assessed | Outcome assessed | Outcome measuring tool | Leisure activity |
|---|---|---|---|---|---|---|---|---|---|
| Gilles et al. 2014 [12] | Belgium | Cross-sectional | 790 | 19.1 | 577 (73.0%) | Sound levels | Attitudes to loud music, attitudes towards a new noise regulation, attitudes towards HPDs | Questionnaire | Nightclub |
| Guéguen et al. 2008 [41] | France | Experimental control randomized trial | 40 | -- | 0 (0%) | Sound levels | Alcohol consumption | Observation by the experimenters | Bars |
| Guéguen et al. 2004 [42] | France | Experimental control randomized trial | 129 | -- | 60 (50%) | Sound levels | Alcohol consumption | Observation by the experimenters | Bars |
| Hunter et al. 2018 [34] | United Kingdom | Case-control retrospective | 28 | 23 | 22 (78.6%) | Sound levels | Attitude to loud music | Focus groups | General leisure activity with amplified music |
| Johnson et al. 2014 [4] | United Kingdom | Cross-sectional | 357 | 21.05 (1.8) | 215 (60.2) | Sound levels | Attitude to loud music | Questionnaire | Nightclub |
| Kayser et al. 2021 [43] | United Kingdom | Cohort prospective | 27 | -- | -- | Emotional connotation (happy vs sad) | Enjoyment, arousal | Questionnaire | Live concert (simulated in a lab) |
| Keppler et al. 2015 [44] | Belgium | Cross-sectional | 163 | 21.2 (2.89) | 127 (77.9%) | Sound levels | Attitude to loud music, attitude to HPDs, use of HPDs, | YANS factor 1, questionnaire, BAHPHL factor 5 | General leisure activity with amplified music |
| Mahomed et al. 2024 [45] | South Africa | Cross-sectional | 462 | -- | 318 (68.8) | Sound levels | Attitude to loud music, effects on hearing health, use of HPD | Questionnaire | Concerts, festivals, Nightclubs, PLD |
| Merrill et al. 2023 [46] | Germany | Experimental control non-randomized trial | 88 | -- | 40 (45.5%) | Music genre | Emotional experience, absorption | Questionnaire | Live concert (simulated in a lab) |
| Sanchez et al. 2019 [47] | Brasil | Cross-sectional | 1822 | 25 (0.9) | 711 (39.3%) | Music genre | Sexual aggression | Questionnaire | Nightclub |
| Swarbrick et al. 2019 [48] | Canada | Experimental control randomized trial | 49 | 31.63 | 22 (44.9%) | Performance type (live vs recorded) | Movement, arousal, emotional valence | Motion capture cap, questionnaire | Live concert and pre-recorded concert (simulated in a lab) |
| ter Bogt & Engels, 2005 [49] | Netherlands | Cross-sectional | 490 | 22.3 (5.03) | 167 (34.0%) | Music genre | MDMA (ecstasy) consumption | Questionnaire | Parties |
| Theorell et al. 2019 [31] | Sweden | Experimental control non-randomized trial | 290 | 45.4 | 211 (72.0%) | Performance type (live vs recorded) | Arousal, emotional valence | VAS | Live concert and pre-recorded concert |
| Tschacher et al. 2023 [50] | Germany | Experimental control non-randomized trial | 89 | 49.8 | 33 (37.3%) | Music genre | Piece-appreciation, piece-connection | Questionnaire | Live concert |
| Warner-Czyz & Cain, 2016 [51] | United States of America | Cross-sectional | 96 | 148 (2.7) | 58 (55.7%) | Sound levels | Attitude to loud music, use of HPDs | YANS factor 1, questionnaire, | General leisure activity with amplified music |
| Weichbold & Zorowka, 2005 [52] | Austria | Cross-sectional | 1213 | 15.7 (1.3) | 608 (50.1%) | Sound levels | Attitude to loud music | Questionnaire | Nightclub |

*(Continued)*

**Table 1.** (Continued)

| First author (& second author), year | Country | Study design | N | Age, Mean (SD) | N female (%) | Musical characteristic assessed | Outcome assessed | Outcome measuring tool | Leisure activity |
|---|---|---|---|---|---|---|---|---|---|
| Weichbold & Zorowka, 2002 [53] | Austria | Cross-sectional | 253 | 15.9 | 174 (69.0%) | Sound levels | Attitude to loud music, use of HPDs | Questionnaire | Nightclub |
| Welch & Fre-maux, 2017 [11] | New Zealand | Cross-sectional | 16 | -- | 9 (56.3%) | Sound levels | Attitude to loud music | Semi-structured interviews | General leisure activity with amplified music |
| Widén, 2013 [54] | Sweden | Cross-sectional | 242 | 17.0 | 108 (44.6%) | Sound levels | Attitude to loud music, use of HPDs, effects on hearing health | YANS-R | General leisure activity with amplified music |
| Widén et al. 2011 [55] | Sweden | Cross-sectional | 543 | -- | 270 (49.7%) | Sound levels | Attitude to loud music | YANS-R | General leisure activity with amplified music |
| Widén & Erlandsson, 2004 [56] | Sweden | Cross-sectional | 1285 | -- | 665 (51.8%) | Sound levels | Attitude to loud music, use of HPDs | YANS factor 1, AHH, HSD | General leisure activity with amplified music |
| Zentner et al. 2008 [57] | Switzer-land | Cross-sectional | 801 | 44.8 (16.5) | 440 (54.9%) | Music genre | Emotional experiences | Questionnaire | Music festival |
| Zocoli et al. 2009 [58] | Brazil | Cross-sectional | 245 | 15.7 | 125 (51.0%) | Sound levels | Attitude to loud music, use of HPD, effects on hearing health | YANS factor 1, questionnaire | General leisure activity with amplified music |

-- not reported.

*median (SD).

†Eichwald et al. only presented the age of their participants in quartiles: 18−32 (25.4%), 33−47 (24.1%), 48−62 (24.8%),>or equal to 63 (25−7%); Forsyth (2012) only presented the age of their participants as % aged under 18 and % aged over 30.

Abbreviations: HPDs: hearing protection devices (HPDs), YANS: Youth Attitude Towards Noise Scale, BAHPHL: Beliefs About Hearing Protection and Hearing Loss, PLD: Personal Listening Devices, VAS: Visual Analog Scale, YANS-R: Youth attitude to noise scale revised (YANS-R), AHH (Adolescent's Habits and use of Hearing protection), HSD (Hearing Symptom Description).

The most common musical characteristic assessed was sound levels as it was investigated in 20 out of 37 studies. Other characteristics investigated include frequency, time stretch, tempo, groove, predictability, emotional connotations, music genre, lyrics, performance style, and performance type. The types of outcomes assessed varied and therefore have been grouped into nine categories:

• Attitude to loud music: including attitudes towards a new noise legislation

• Movement

• Groove

• Body feel

• Emotions: including enjoyment, absorption, piece appreciation, piece connection, emotions like happiness and calm-ness which are defined as valence, arousal, emotional experiences, emotional intensity

• Harmful behavior: including substance use such as alcohol consumption and drug consumption, sexual tension and aggression, aggression incidents

- Effects on hearing health

- Attitude to hearing protection devices (HPDs)

- Use of HPDs

Nightclubs and concerts were the most common domains of focus in the included studies (18 out of 37 studies). For lab-based studies, such as the one by Cameron et al. [32], which took place in a research performance hall (LiveLab), the leisure activity being simulated was reported in Table 1. We did not specifically note in Table 1 that these studies were conducted in lab settings, as their high ecological validity justified reporting the intended leisure activity rather than the research environment. Twelve studies out of 37 did not focus on a specific leisure activity as their research question included any leisure activity where the purpose of the attendance was amplified music [29,30,33,34]. Dotov et al. [15] study design includes four intervention groups (low groove/low tempo, high groove/low tempo, low groove/high tempo, and high groove/high tempo).

**Risk of bias**

Both cohort studies (Table 2) were considered having a low risk of bias. Conversely, the case-control studies (Table 3) by Carter & Black [35] and Hunter et al. [34] were determined to have a high risk of bias due to the exposure and comparability criteria respectively. Out of the 7 points that could be rewarded for the risk of bias analysis of cross-sectional studies (Table 4), the included papers' scores ranged from 1 to 5. The lowest score was assigned to Forsyth [40], receiving only one point for case representation. This was largely due to the study design, which relied on observers taking field notes to record outcomes. These subjective observations significantly increased the risk of bias. In contrast, Sanchez et al. [47] and Mahomed et al. [45] had the most limited risk of bias, scoring five points, primarily due to the selection criteria.

Experimental control trials were split based on their randomization procedure. None of the 8 randomized control trials (Table 5), which also included crossover studies, were judged to be of low risk. Dolan et al. [37] and Engels et al. 2011 [39] were both considered to have a high risk of bias. Despite Dolan et al. [37] exhibiting a low risk of bias for most criteria, their randomization procedure was flagged as high risk, which automatically classified the study as having a high risk of bias. The included non-randomized control trials (Table 6) also had a high risk of bias. Notably, the study by Theorell et al. [31] was deemed to have a critical risk of bias, primarily due to inadequate control of confounding factors and flawed participant selection. They chose cohorts from different populations, with the pre-recorded group consisting solely of elderly listeners aged 63 and above, while the live performance cohort had a broader age range of 22–83.

**Table 2. Risk of bias NOS for cohort studies.**

| Study ID | Selection | Comparability | Outcome | Total score |
|---|---|---|---|---|
| Beach et al. 2019 [2] | ++++ | ++ | +++ | 9 |
| Kayser et al. 2021 [43] | -+++ | +- | ++- | 6 |

**Table 3. Risk of bias NOS for case-control studies.**

| Study ID | Selection | Comparability | Exposure | Total score |
|---|---|---|---|---|
| Carter & Black 2017 [35] | --++ | +- | -+- | 4 |
| Hunter et al. 2018 [34] | --++ | -- | -++ | 4 |

**Table 4. Risk of bias NOS for cross-sectional studies.**

| Study ID | Selection | Comparability | Outcome | Total score |
|---|---|---|---|---|
| Degeest et al. 2021 [33] | +-+ | **+-** | -+ | 4 |
| Degeest et al. 2018 [29] | --+ | **+-** | -+ | 3 |
| Eichwald et al. 2022 [30] | ++- | **+-** | -+ | 4 |
| Forsyth (2009) [40] | +-- | -- | -- | 1 |
| Gilles et al. 2014 [12] | -+- | **++** | -+ | 4 |
| Johnson et al. 2014 [4] | +-- | -- | -+ | 2 |
| Keppler et al. 2015 [44] | -+- | **+-** | -+ | 3 |
| Mahomed et al. 2024 [45] | +++ | **-+** | -+ | 5 |
| Sanchez et al. 2019 [47] | +++ | **+-** | -+ | 5 |
| ter Bogt & Engels (2005) [49] | +-- | -- | -+ | 2 |
| Warner-Czyz & Cain (2016) [51] | +-- | +- | -+ | 3 |
| Weichbold & Zorowka (2005) [52] | ++- | -- | -+ | 3 |
| Weichbold & Zorowka (2002) [53] | +-- | -- | -+ | 2 |
| Welch & Fremaux (2017) [11] | +-- | -- | -+ | 2 |
| Widén (2013) [54] | ++- | +- | -+ | 4 |
| Widén et al. 2011 [55] | ++- | +- | -+ | 4 |
| Widén & Erlandsson (2004) [56] | +-- | -- | -+ | 2 |
| Zentner et al. 2008 [57] | +-- | -- | -+ | 2 |
| Zocoli et al. 2009 [58] | -+- | +- | -+ | 3 |

**Table 5. Risk of bias RoB2 for randomized (crossover) trials.**

| Study ID | Randomization process | Period and carryover effects* | Deviations from the intended interventions | Missing outcome data | Measurement of outcome | Selection of the reported results | Overall |
|---|---|---|---|---|---|---|---|
| Burger et al. 2018 [14] | – | +/- | +/- | – | +/- | +/- | +/- |
| Dolan et al. 2018 [37] | + | +/- | – | – | – | – | + |
| Dotov et al. 2021 [15] | – | – | – | – | +/- | +/- | +/- |
| Engels et al. 2012 [38] | – | – | – | +/- | +/- | – | +/- |
| Engels et al. 2011 [39] | – | +/- | +/- | + | + | +/- | + |
| Guéguen et al. 2008 [41] | +/- | NA | +/- | – | +/- | – | +/- |
| Guéguen et al. 2004 [42,48] | +/- | NA | +/- | – | +/- | – | +/- |
| Swarbrick et al. 2019 [48] | – | NA | – | +/- | +/- | +/- | +/- |

+Low risk, +/- Some concerns, – High risk, NA Not applicable

*only for crossover studies

**Table 6. Risk of bias ROBINS-I for non-randomized trials.**

| Study ID | Confounding | Selection of participants into the study | Classification of interventions | Deviations from intended interventions | Missing data | Measurement of outcomes | Selection of the reported result | Overall |
|---|---|---|---|---|---|---|---|---|
| Cameron et al. 2022 [32] | Moderate | Moderate | Low | Low | Serious | Moderate | Low | Serious |
| Coutinho & Scherer (2017) [36] | Moderate | Moderate | Low | No information | Low | Serious | Low | Serious |
| Egermann et al. 2013 [17] | Moderate | Low | Moderate | Low | Low | Low | Low | Moderate |
| Merill et al. 2023 [46] | Serious | Low | Moderate | Low | Moderate | Low | Moderate | Serious |
| Theorell et al. 2019 [31] | Critical | Serious | Moderate | No information | Serious | Low | Moderate | Critical |
| Tschacher et al. 2023 [50] | Low | Low | Low | Low | Moderate | Low | Low | Moderate |

## Effects on attitude to loud music

Results across studies showed different attitudes towards loud music. As presented in Table 7, 75.9% of outcomes investigated reported that less than 50% of participants had a positive attitude towards loud music. Beach et al. [2] is one of the few studies which recorded particularly high percentages of positive attitude as 76.1% (n = 422) and 76.6% (n = 290) of nightclubs and live music venues attendees respectively have reported not avoiding particular nightclubs or venues which played extensively loud music. Furthermore, Gilles et al. [12] determined that in two of their research conditions 75.6% and 64.6% of their participants believed noise levels should stay the same or be raised. In the study of Cameron et al. [32] on a scale from 1 (much quieter) to 9 (much louder) a mean of 6.18 (SD = 1.59, 95% CI = 5.74–6.62) was reported. The Youth Attitude to Noise Scale (YANS) is a 19-item questionnaire measuring attitudes toward noise on a 5-point Likert scale that has been developed by Olsen & Erlandsson in an unpublised dissertation. These items group into four factors: noise linked to youth culture, ability to concentrate in noise, daily noises, and influence over the sound environment. The full questionnaire has been extracted from another article by Zocoli et al. [59] and is available in S3 File. This review focuses only on the first factor, which relates to leisure activities involving loud music. The Youth attitude to noise scale revised (YANS-R) is a revised version based on the first factor of the original YANS [55]. It includes 11 items targeting attitudes toward loud music in specific settings. These questionnaires were used in 8 studies to assess participants' sentiments towards noise and its association to elements of youth culture. Among these 8 studies, all except Zocoli et al. [58] reported statistically significant results, with scores ranging from 2.27 to 3.46, as shown in Table 7.

Carter & Black [35] described that a considerable reason for people to avoid loud situations at leisure activities is that it is "too hard to hear conversation". Contrarily, Welch & Fremaux [11] interviewed patrons to understand why they enjoyed loud sounds, and they found the most important reasons to be: arousal through "enhancing emotions, motivation to move and providing direct physical sensations". Hunter et al. [34] collected qualitative data where individuals attending leisure activities explained both points of view: "The benefits outweigh the risks listening to it at a certain volume, it definitely would compromise the experience having to turn it down", "When you can feel your body shaking because of the bass, it's too much".

## Effects on movement

Four of the 34 included studies presented data about the effects of music on participants' movements. Researched musical characteristics were different in each of the studies. Cameron et al. [32] investigated the effects of low frequencies using very low frequency (VLF) speakers on the participants' movement speed and self-reported movements. There was a speed difference of +0.118 m/s (p < 0.001) when VLF was turned on compared to when it was turned off. On a scale from 1 (not at all) to 9 (very much) the mean movement rating of the overall concert was 5.24 (p < 0.001). Dotov et al. [15] researched the effects of groove and tempo on four measures of movement. The difference of each participant's measures, when the independent variable (groove or tempo) was high vs low, were calculated, as seen in Table 8. In this review, we used the low groove/low tempo group as the control, representing the low condition for both variables. Only movement energy showed a significant increase, with changes of +68.0 $kgcm^2s^{-2}$ (p < 0.0001) when high-groove music was played, and +35.4 $kgcm^2s^{-2}$ (p < 0.0001) with high-tempo music. Swarbrick et al. [48] investigated the impact of performance type on the vigor and entertainment of the participants' movements. Concerning vigor, the live and pre-recorded conditions means were 16.9 mm/s (95% CI:12.6 to 21.3) and 8.33 mms/s (95% CI: 5.72 to 10.9) respectively. This significant difference was not observed for entertainment. Lastly, Burger et al. [14] researched the impact of frequency flux, time stretch, and tempo on the synchronization ability of several body parts (foot, hip, hand, and head) to the bar and beat. Due to the extensive data, the detailed results were not included in Table 8. Overall, their results revealed a complex interplay among all three musical characteristics. For instance: "strong low-frequency spectral flux was found to result in tighter synchronization at slower tempi at the beat level, whereas it became a less salient cue at faster tempi" [14].

**Table 7. Outcome attitude to loud music.**

| First author, year | Outcome[†], measuring tool | Subgroups | N | The percentage for positive attitude[‡] | Mean[§] (SD) | Range | 95% CI | p-value t-test[ǀ] |
|---|---|---|---|---|---|---|---|---|
| Beach et al. 2019 [2] | "I like my live music to be loud- the louder the better. I'm there for a good time, I'm not thinking about my ears, my health, or anything else" | Nightclubs | 555 | 55.1% | | | | |
| | | Live music venues | 378 | 22.9% | | | | |
| | "Do you find that the music at most of the nightclubs/live music venues you go to is usually *just right*" or "*not as loud as you would like*" | Nightclubs | 555 | 14.2% | | | | |
| | | Live music venues | 378 | 20.2% | | | | |
| | "I do not avoid particular night-clubs/live music venues that I know play music too loud" | Nightclubs | 555 | 76.1% | | | | |
| | | Live music venues | 378 | 76.6% | | | | |
| | Participants who disagree with this statement: "When I go out, I want to chat with my friends as well as dance so I'd prefer if there were some quieter places to sit and chat when we're taking a break" | Nightclubs | 555 | 14.4% | | | | |
| | Participants who disagree with this statement: "When I go out to a live music venue, I want to chat with my friends as well as enjoy the music so I'd prefer if the noise levels were lower" | Live music venues | 378 | 62.8% | | | | |
| Cameron et al. 2022 [32] | Preference for loud music: *Scale from 1 (much quieter) to 9 (much louder) with 5=no change* | -- | 50 | | 6.18 (1.59) | | 5.74-6.62 | <0.001[*] |
| Carter & Black 2017 [35] | "Do you prefer to avoid some places (e.g., clubs, dance parties), or activities (e.g., motor sports) because they are too loud?": *No* | -- | 210 | 48.1% | | | | |
| | "Have you ever left a place, or stopped doing an activity, because it was too loud?": *Never* | -- | 210 | 37.6% | | | | |
| Degeest et al. 2021 [33] | YANS factor 1: *scale from 1 to 5[¶]* | -- | 242 | --- | 2.9 (0.64) | 1.38-4.75 | 2.82-2.98 | 0.0158[*] |
| Degeest et al 2018 [29] | YANS factor 1: *scale from 1 to 5[¶]* | -- | 43 | -- | 2.27 (0.61) | 1.25-3.50 | 2.09-2.45 | <0.001[*] |
| Eichwald et al. 2023 [30] | "Sound levels at venues of events should be limited to reduce the risk of hearing loss": *Disagree or strongly disagree* | -- | 3526 | 16%[#] | | | | |

*(Continued)*

**Table 7.** (Continued)

| First author, year | Outcome†, measuring tool | Subgroups | N | The percentage for positive attitude‡ | Mean§ (SD) | Range | 95% CI | p-value t-testᴵ |
|---|---|---|---|---|---|---|---|---|
| Gilles et al. 2014 [12] | Negative responses to a new noise legislation in Flanders, Belgium, which aims to control recreational noise | Research 1: prior to the new noise legislation | 41 | 19.5% | | | | |
| | | Research 2: after the new noise legislation | 749 | 18.0% | | | | |
| | Party behavior after more strict regulation: participants who responded that they would go somewhere else where the noise levels are higher | Research 1: prior to the new noise legislation | 41 | 4.90% | | | | |
| | | Research 2: after the new noise legislation- at youth clubs, party halls, and discotheques | 749 | 5.5% | | | | |
| | | Research 2: after the new noise legislation- at festivals/open air activities | 749 | 7.9% | | | | |
| | Opinions that discotheque levels should be raised or stay the same | Research 1: prior to the new noise legislation | 41 | 75.6% | | | | |
| | | Research 2: after the new noise legislation- at youth clubs, party halls, and discotheques | 749 | 44.7% | | | | |
| | | Research 2: after the new noise legislation- at festivals/open air activities | 749 | 64.6% | | | | |
| | Music levels of <100dB are too quiet | Research 1: prior to the new noise legislation | 41 | 2.40% | | | | |
| | Music levels of>100dB are perfect or too quiet | Research 1: prior to the new noise legislation | 41 | 49.0% | | | | |
| Johnson et al. 2014 [4] | Noise levels in nightclubs should not be limited to a volume that is not damaging | -- | 332 | 29.8% | | | | |
| Keppler et al. 2015 [44] | YANS factor 1: *scale from 1 to 5*¶ | -- | 163 | | 2.44 (0.64) | 1.13-4.38 | 2.34-2-54 | <0.001* |
| Mahomed et al. 2024 [45] | Music level rating: *the sound levels are low or too low* | Concerts/festivals | 390 | 2.56% | | | | |
| | | Nightclubs | 343 | 2.92% | | | | |
| | Preferred volume of PLDs: *maximum volume* | -- | 436 | 30.7% | | | | |
| | The participant would most likely do nothing if the music is too loud | -- | 453 | 30.9% | | | | |
| Warner-Czyz & Cain (2016) [51] | YANS factor 1: *scale from 1 to 5*¶ | -- | 96 | | 3.40 (0.80) | | 3.24-3-56 | <0.001* |

*(Continued)*

**Table 7.** (Continued)

| First author, year | Outcome†, measuring tool | Subgroups | N | The percentage for positive attitude‡ | Mean§ (SD) | Range | 95% CI | p-value t-testˡ |
|---|---|---|---|---|---|---|---|---|
| Weichbold & Zorowka (2005) [52] | "What is your opinion on the volume in nightclubs?: it *should be louder*, or *it is fine as it is*" | -- | 1213 | 56.2% | | | | |
| | "Suppose the music in nightclubs was played a bit quieter than before. How often would you then go to the nightclub compared to now?: *less often,* or *I would go to another club where the music is louder*" | -- | 1213 | 5.4% | | | | |
| Weichbold & Zorowka (2002) [53] | "Do you find the music in discos too loud?: *Never*" | -- | 215 | 35.5% | | | | |
| Widén (2013) [54] | YANS-R: *scale from 1 to 5*¶ | -- | 242 | | 3.46 (0.627) | | 3.38-3-54 | <0.001* |
| Widén et al. 2011 [55] | YANS-R: *scale from 1 to 5*¶ | -- | 543 | | 3.33 (0.630) | | 3.28-3.38 | <0.001* |
| Widén & Erlandsson (2004) [56] | YANS factor 1: *scale from 1 to 5*¶ | -- | 1285 | | 2.86 (0.920) | | 2.81-2.91 | <0.001* |
| Zocoli et al. 2009 [58] | YANS factor 1: *scale from 1 to 5*¶ | -- | 245 | | 2.94 (1.33) | | 2.77-3.11 | 0.48 |

*Statistically significant.

†Outcomes of certain studies were adjusted to conform to a positive attitude toward loud music.

‡Defined as an attitude where loud noise is seen as unproblematic and the preferred volume of music at leisure activities.

§MDs have a + or – sign in front of them.

ˡThe null hypotheses (value of 5 for Cameron et al. [32] and value of 3 for all studies using the YANS tool) were used to compare differences in the one-sample t-test.

¶A higher score indicates an attitude where noise is seen as unproblematic.

#*Result estimated from a figure in the paper as no specific value was given in-text.*

## Effects on emotions

From the eleven studies which investigated the effects of music on emotions, eight sub-outcomes could be identified. Most of the articles can be seen in Table 9. Enjoyment was the first sub-outcome explored. It was measured by Cameron et al. [32], Kayser et al. [43], and Egermann et al. [17] who used frequency, emotional connotations, and expectedness as their independent variables respectively. The VLF effects on in-concert enjoyment was +0.0741 (p>0.05). Yet, the overall enjoyment post-concert was significantly higher than 5 (indicating neutrality) (M=6.57, p<0.0001). Songs with happy connotations increased enjoyment by 1.44 (p>0.05), failing to reach statistical significance. Egermann et al. [17] could not provide access to raw data; however, they reported no effect on enjoyment ratings for both very unexpected and very expected segments.

Absorption, which included both engagement and dissociation, was explored by Merrill et al. [46]. The authors conducted additional statistical analyses (not presented in Table 9) and found that the romantic music genre elicited significantly higher dissociation ratings compared to the contemporary genre. However, no significant effects were observed for engagement.

**Table 8. Outcome movement.**

| First author, year | Movement measuring tool | Type of movement | | Intervention | | N | Mean† units (SD) | 95% CI (low to high) | P-value for t-test |
|---|---|---|---|---|---|---|---|---|---|
| Cameron et al 2022 [32] | Motion capture headband | General movement speed | | Frequency (VLF on –vs VLF off) | | 43 | +0.118‡ m/s (0.115) | 0.0743 to 0.161 | <0.001* |
| | Questionnaire | Overall self-reported movement. (1 = not at all, 9 = very much) | | | | 43 | 5.24 (2.34) | 4.59 to 5.88 | <0.001* |
| Dotov et al. 2021 [15] | Motion capture cap | Tempo alignment§ | PC1 | Groove (high –low) | | 33 | −0.0224 bpm (0.129) | −0.0656 to 0.0209 | 0.326 |
| | | | | Tempo (high – low) | | 33 | +0.0156 bpm (0.0956) | −0.0171 to 0.0484 | 0.364 |
| | | | PC2 | Groove (high –low) | | 33 | −0.0403 bpm (0.134) | −0.0860 to 0.00538 | 0.0983 |
| | | | | Tempo (high – low) | | 33 | +0.0025 bpm (0.131) | −0.0422 to 0.0473 | 0.914 |
| | | Amplitude | | Groove (high –low) | | 33 | +0.136 cm (6.27) | −2.00 to 2.28 | 0.903 |
| | | | | Tempo (high – low) | | 33 | −0.759 cm (5.45) | −2.62 to 1.10 | 0.436 |
| | | Energy | | Groove (high –low) | | 33 | +68.0 kgcm$^2$s$^{-2}$ (53.1) | 49.1 to 86.1 | <0.001* |
| | | | | Tempo (high – low) | | 33 | +35.4 kgcm$^2$s$^{-2}$ (37.1) | 22.7 to 48.8 | <0.001* |
| | | Dimensionality | | Groove (high –low) | | 33 | +0.0139 (0.216) | −0.0599 to 0.0878 | 0.718 |
| | | | | Tempo (high – low) | | 33 | −0.0330 (0.242) | −0.116 to 0.0487 | 0.434 |
| Swarbrick et al. 2019 [48] | Motion capture caps | Entertainment (ranges between 0.0 = no entrainment and 1.0 = perfect entrainment) | | Performance type | Live performance | 24 | 0.0722 (0.104) | 0.0306 to 0.114 | -- |
| | | | | | Pre-recorded performance | 25 | 0.0596 (0.0875) | 0.0253 to 0.0939 | |
| | | Vigor | | | Live performance | 24 | 16.9 mm/s (11.0) | 12.6 to 21.3 | -- |
| | | | | | Pre-recorded performance | 25 | 8.33 mms/s (6.62) | 5.72 to 10.9 | |

*Statistically significant

†MDs have a + or – sign in front of them

‡Mean was normalized

§Measured on two axes (PC1 and PC2).

Abbreviations: VLF: very low frequency, PC1: first principal component of movement dimensions, PC2: second principal component of movement dimensions

Piece appreciation and connection were investigated across different music genres by Tschacher et al. [50] using a multilevel regression model. Their findings indicated that participants rated their appreciation for the piece significantly higher when the genre was classical or romantic compared to contemporary music (*p* < 0.01). However, no significant differences were found for piece connection [50]. These results were excluded from Table 9 due to the complexity of the regression model.

**Table 9. Outcome emotions.**

| First author, year | Emotion measuring tool | Emotion sub-outcome measured: *scale* | Intervention | | N | Mean† (SD) | 95% CI (low to high) | P-value‡ |
|---|---|---|---|---|---|---|---|---|
| Cameron et al 2022 [32] | Text messages | In concert enjoyment: *1 (ok) to 9 (extremely enjoyable)* | Frequency (VLF on – VLF off) | | 21 | +0.0741§ (0.258) | −0.0362 to 0.184 | 0.214 |
| | Post-performance questionnaire | Enjoyment: *1 (ok) to 9 (extremely enjoyable)* | | | 51 | 6.57 (2.12) | 6.00 to 7.25 | <0.001* |
| Dolan et al 2018 [37] | Verbal questionnaire | Emotional intensity: *0 (not all all/none) to 5(totally/ completely)* | Performance style (improvised- prepared) | | 22 | +1.32 (1.58) | 0.66 to 1.98 | <0.001* |
| Dotov et al. 2021 [15] | Questionnaire | Happiness: *0 (very sad) to 5 (very happy)* | Groove (high –low) | | 33 | +0.636 (0.948) | 0.313 to 0.960 | <0.001* |
| | | | Tempo (high - low) | | 33 | +0.606 (0.851) | 0.316 to 0.896 | <0.001* |
| | | Emotion intensity: *0 (not intense) to 5 (very intense)* | Groove (high – low) | | 33 | +1.73 (1.54) | 1.20 to 2.25 | <0.001* |
| | | | Tempo (high – low) | | 33 | +0.788 (1.12) | 0.405 to 1.17 | <0.001* |
| Kayser et al. 2021 [43]‖ | Questionnaire | Enjoyment: *−5 (unpleasant) to +5 (pleasant)* | Emotional connotation (happy-sad) | | 27 | +1.44 (2.83) | −0.00132 to 2.89 | 0.0657 |
| | | Arousal: *−5 (calm) to +5 (excited)* | | | 27 | +2.89 (3.13) | 1.71 to 4.07 | <0.001* |
| Merrill et al. 2023 [46] | Questionnaire | Positive emotions: *linear mixed model with emmeans where higher values indicate a higher intensity* | Music genre | Classical | 88 | +0.233 | −0.107 to 0.574 | -- |
| | | | | Contemporary | 88 | −0.321 | −0.661 to 0.0187 | |
| | | | | Romantic | 88 | +0.0479 | −0.292 to 0.388 | |
| | | Negative emotions: *linear mixed model with emmeans where higher values indicate a higher intensity* | | Classical | 88 | −0.370 | −0.562 to −0.178 | -- |
| | | | | Contemporary | 88 | +0.739 | 0.548 to 0.931 | |
| | | | | Romantic | 88 | −0.442 | −0.634 to −0.251 | |
| | | Mixed emotions: *linear mixed model with emmeans where higher values indicate a higher intensity* | | Classical | 88 | +0.107 | −0.304 to 0.517 | -- |
| | | | | Contemporary | 88 | −0.468 | −0.878 to 0.0575 | |
| | | | | Romantic | 88 | +0.393 | −0.0174 to 0.803 | |
| | | Engagement: *linear mixed model with emmeans where higher values indicate a higher intensity* | | Classical | 88 | +0.0935 | −0.148 to 0.335 | -- |
| | | | | Contemporary | 88 | −0.0905 | −0.331 to 0.150 | |
| | | | | Romantic | 88 | +0.0204 | −0.221 to 0.261 | |
| | | Dissociation: *linear mixed model with emmeans where higher values indicate a higher intensity* | | Classical | 88 | −0.0244 | −0.180 to 0.131 | -- |
| | | | | Contemporary | 88 | −0.107 | −0.261 to 0.0476 | |
| | | | | Romantic | 88 | 0.125 | −0.0298 to 0.280 | |

*(Continued)*

| First author, year | Emotion measuring tool | Emotion sub-outcome measured: *scale* | Intervention | | N | Mean† (SD) | 95% CI (low to high) | P-value‡ |
|---|---|---|---|---|---|---|---|---|
| Theorell et al. 2019 [31] | VAS | Arousal: *0–10* | Live performance | Experiment I | 23 | 7.1 (2.5) | 6.08 to 8.12 | 0.113 |
| | | | | Experiment II | 63 | 6.8 (2.9) | 6.08 to 7.52 | <0.001* |
| | | | Pre-recorded | Experiment I | 92 | 4.7 (2.3) | 4.23 to 5.17 | 0.001* |
| | | | | Experiment II | 112 | 5.6 (2.3) | 5.17 to 6.03 | <0.001* |
| | | Happiness: *0–10* | Live performance | Experiment I | 23 | 8.6 (1.1) | 8.15 to 9.05 | <0.001* |
| | | | | Experiment II | 63 | 8.0 (1.8) | 7.56 to 8.44 | 0.093 |
| | | | Pre-recorded | Experiment I | 92 | 5.2 (2.3) | 4.73 to 6.67 | 0.003* |
| | | | | Experiment II | 112 | 5.6 (2.1) | 5.21 to 5.99 | <0.001* |
| | | Calmness: *0–10* | Live performance | Experiment I | 23 | 8.3 (1.4) | 7.73 to 8–87 | 0.001* |
| | | | | Experiment II | 63 | 8.0 (1.9) | 7.53 to 8.47 | 0.343 |
| | | | Pre-recorded | Experiment I | 92 | 6.0 (2.4) | 5.51 to 6.49 | 0.001* |
| | | | | Experiment II | 112 | 6.2 (2.4) | 5.76 to 6.64 | <0.001* |

*Statistically significant.

†MDs have a + or – sign in front of them. Thes estimated marginal means of the linear mixed models from Merrill et al. [46] also have a + or – sign in front of them.

‡All p values are from paired t-tests except for Theorrell et al. [31] since their raw data could not be retrieved so the Wilcoxon test values from the article were used.

§Normalized.

‖Kayser et al. [43] repeated their experiments for each of their intervention variables (happy and sad songs). Hence the mean of the two values recorded for each variable was calculated for each participant before finding their MD.

Emotional valence, including happiness and calmness, was addressed in three studies as seen in Table 9. For example, Dotov et al. [15] demonstrated that increasing the groove or tempo increased the participants' sense of happiness by 0.636 (p < 0.001) and 0.606 (p < 0.001) respectively on a 6-point Likert scale.

Arousal was explored by Kasyer et al. [43] who found an increase of 2.89 (p < 0.001) on an 11-point Likert scale in songs with happy connotations. Egermann et al. [17] results' showed that unexpected events had a significant impact on arousal. Theorell et al. [31] also collected data on arousal and valence; however, due methodoligical issues and potential bias, the significance of these results is very limited. The study by Swarbrick et al. [48] also measured these sub-outcomes, however, due to minimal difference in scores between the post and pre-concert questionnaires for these sub-outcomes, their data was not reported in Table 9.

Other emotional experiences were investigated by Zentner et al. [57], Coutinho et al. [36], and Merrill et al. [46]. Contemporary music had an estimated marginal mean (emmeans) of +0.739 (95% CI = 0.548–0.931) for negative emotions, which was significantly higher than the emmeans of −0.370 (95% CI: −0.562 to −0.178) and −0.442 (95% CI: −0.634 to −0.251) for classical and romantic pieces respectively [46]. Due to practical reasons and lack of raw data, neither of the data for the articles by Zentner al. [57] and Coutinho et al. [36] could be extracted. Zentner et al. [57] found that the most felt emotions at a classical, jazz, rock, and world genre music festival were relaxed, happy, joyful, and dreamy. Coutinho et al. [36] reported that only feelings of wonder, sadness, and boredom were statistically different between their two live versus audio-video-recording performances. Wonder and sadness were higher in the live condition whilst boredom was lower.

Emotional intensity was investigated by Dolan et al. [37] who recorded an increase of +1.32 (p < 0.001) on a scale from 0 (not all all/none) to 5 (totally/completely) when the performance was improvised compared to prepared. Dotov et al. [15] effects of groove and tempo on emotional intensity was of +1.73 (p < 0.0001) and +0.788 (p < 0.001) respectively.

## Effects on harmful behavior

A total of 7 articles reported data on harmful behavior. The most common musical characteristic researched was music genre. Surprisingly, the ANCOVA test performed by Engels et al. [38] revealed that classical music significantly increased overall alcoholic consumption compared to the three other genres (popular, hard rock, and gangsta rap). On the other hand, Forsyth [40] recorded their highest percentage of drunk clients (78.3%) at hardcore venues, although classical music was not investigated. Hardcore venues also had the highest recorded number of aggression incidents (n: 19 out of the 487 clients), although this is hard to compare as each venue had a different number of total clients. Other results by Forsyth [40] can be seen in Table 10. Hardcore venues showed a high percentage of MDMA use (48%) especially compared to the visitors of the club/mellow party [49]. Using a multilevel regression model, Sanchez et al. [47] found that attending a nightclub playing funk, electronic, pop dance, or forro/zouk significantly increased the odds of experiencing sexual assault compared to nightclubs playing eclectic music.

The two studies by Guéguen et al. [41,42] investigated the impact of sound levels on alcohol consumption. In both cases high sound levels had a main effect on the number of drinks ordered (p < 0.03 and p < 0.001 respecitively). Finally, a study by Engels et al. [39] investigated whether alcohol references in song lyrics would have an impact on alcohol consumption. They found that the bars that played songs with alcohol references had a mean turnover €8 higher than those without.

## Hearing and HPD outcomes

Some of the included studies collected data on hearing health outcomes. For instance, Beach et al. [2] and Degeest et al. [33] reported that 86.0% and 64.8% of their participants, respectively, had experienced noise-induced tinnitus. Similar findings were observed for NIHL: 70.8% of participants in Degeest et al. [33] reported experiencing NIHL "sometimes to always" after noise exposure. Mahomed et al. [45] also found that 21.4% of their participants experienced NIHL after each loud noise exposure. Additionaly, 82.6% of participants described being at least sometimes sensitive to noise following loud music exposure [33]. Widen [54] reported that 5.4% of their 240 participants had permanent tinnitus and 14.1% experienced hyperacusis at least 50% of the time. Zocoli et al. [58] found that participants commonly experienced temporary tinnitus after specific music-related exposures: 45% after leaving a disco club, 28% after attending a concert, and 11% after listening to music through an audio device. However, permanent tinnitus was also rare in their sample, reported by only 0.4% of participants. [59]

Participants' attitudes toward HPDs was another recurrent outcome investigated by the included studies. Beach et al. [2], Degeest et al. [33] and Eichwald et al. [30] reported a variety in willingness to use HPDs ranging from 14.9% to 68.3%. Degeest et al. [29,33] used factor 5 of the Beliefs About Hearing Protection and Hearing Loss (BAHPHL) [60] questionnaire to assess the behavioral intentions of their participants regarding hearing health on a scale from 1–5. A high score corresponds to an attitude where one does not care about the possible consequences of hearing loss and is unaware of the benefits of HPDs. Whilst the Degeest et al. 2021 study [33] reported a mean of 3.3 (95% CI = 3.03–3.57) on the questionnaire, the Degeest et al. 2018 study [29] showed a lower mean value of 1.98 (95% CI = 1.66–2.30). This may be due to differences in sample sizes (n = 236 and n = 43 respectively) or the different age groups of focus (teenagers attending high school and young adults of 18–30 years old respectively). Keppler et al. [44] reported a mean of 2.94 (SD = 1.10) on the BAHPHL questionnaire. Prevention of noise-induced hearing symptoms was determined as the most important reason to use HPD by Gilles et al. [12]. The most significant reason to not use it was" I never thought about using it."

Carter & Black [35] reported that use of hearing protectors in loud environments was low among all participants. HPD use was most frequently reported during several of the highest-noise activities (nightclubbing, firearms, and power tool use). However all articles reported a low use of HPD's with the highest percentage reaching 17% [44,45,51,53,54,56,58].

**Table 10.  Outcome harmful behavior.**

| First author, year | Outcome measuring tool | Type of harmful behavior | Harmful behavior sub-outcome measure: *scale* | Intervention | | N | Events/ number | Percent-age | Mean (SD) | 95% CI (low to high) | p-value[†] |
|---|---|---|---|---|---|---|---|---|---|---|---|
| Engels et al. 2012 [38] | Observations | Alcohol con-sumptions | Grams of alcohol consumed per participant | Music genre | Popular | 52 | -- | -- | 36.24 (15.5) | 32.0 to 40.5 | -- |
| | | | | | Hard rock | 70 | -- | -- | 35.6 (15.1) | 32.0 to 39.1 | -- |
| | | | | | Classical | 63 | -- | -- | 46.3 (15.1) | 42.6 to 50.1 | -- |
| | | | | | Gangsta rap | 64 | -- | -- | 39.9 (15.2) | 36.1 to 43.6 | -- |
| | | | Number of drinks consumed per participant | Music genre | Popular | 52 | -- | -- | 4.72 (0.230) | 4.66 to 4.78 | -- |
| | | | | | Hard rock | 70 | -- | -- | 4.43 (0.190) | 4.39 to 4.47 | -- |
| | | | | | Classical | 63 | -- | -- | 5.23 (0.200) | 5.18 to 5.28 | -- |
| | | | | | Gangsta rap | 64 | -- | -- | 4.68 (0.200) | 4.63 to 4.73 | -- |
| Engels et al. 2011 [39] | -- | Alcohol consumption | Quantity of euros spent on alcoholic beverages during the session | Song lyrics | With alcohol references | -- | -- | -- | 191 | -- | -- |
| | | | | | Without alcohol references | -- | -- | -- | 183 | -- | -- |
| Forsyth (2009) [40] | Observations | Alcohol consumption | Number of drunk clients | Music genre | Hardcore venues | 487 | -- | 78.3% | -- | -- | -- |
| | | | | | Cheesy-pop venues | 500 | | 63.4% | | | |
| | | | | | Urban venues | 455 | | 66.2% | | | |
| | | | | | Mixed venues | 251 | | 58.1% | | | |
| | | | | | Old-school rave | 154 | | 38.8% | | | |
| | | Drug consumption | Number of clients obviously on drugs | | Hardcore venues | 487 | -- | 34.8% | -- | -- | -- |
| | | | | | Cheesy-pop venues | 500 | | 2.90% | | | |
| | | | | | Urban venues | 455 | | 5.80% | | | |
| | | | | | Mixed venues | 251 | | 4.40% | | | |
| | | | | | Old-school rave | 154 | | 50.0% | | | |
| | | Sexual tension: *0 (low)- 27 (high)* | | | Hardcore venues | 487 | 15.9 | -- | -- | -- | -- |
| | | | | | Cheesy-pop venues | 500 | 12.7 | | | | |
| | | | | | Urban venues | 455 | 15.0 | | | | |
| | | | | | Mixed venues | 251 | 17.0 | | | | |
| | | | | | Old-school rave | 154 | 6.10 | | | | |
| | | Aggression incidents | | | Hardcore venues | 487 | 19 | -- | -- | -- | -- |
| | | | | | Cheesy-pop venues | 500 | 8 | | | | |
| | | | | | Urban venues | 455 | 7 | | | | |
| | | | | | Mixed venues | 251 | 0 | | | | |
| | | | | | Old-school rave | 154 | 0 | | | | |

*(Continued)*

**Table 10.** (Continued)

| First author, year | Outcome measuring tool | Type of harmful behavior | Harmful behavior sub-outcome measure: *scale* | Interven- tion | | N | Events/ number | Percent- age | Mean (SD) | 95% CI (low to high) | p-value† |
|---|---|---|---|---|---|---|---|---|---|---|---|
| Guéguen et al. 2008 [41] | Observations | Alcohol consumption | Number of drinks ordered per participant | Sound levels | High sound level (88dB) | 40 | -- | -- | 3.40 (0.990) | -- | <0.03* |
| | | | | | Usual sound level (72dB) | | | | 2.60 (1.14) | | |
| Guéguen et al. 2004 [42] | Observations | Alcohol con- sumptions | Number of drinks ordered per participant | Sound levels | High sound lev- els (88-91dB) | 60 | -- | -- | 3.70 | -- | <0.001* |
| | | | | | Usual sound level (72-75dB) | 60 | | | 2.60 | | |

*Statistically significant

†The p values for the Guéguen et al. 2008 [41] and 2004 [42] articles are taken from theirunpaired t-test and analysis fo varience respectively.

## Discussion/conclusion

We explored the effects of several musical characteristics on outcomes regarding participants' experience or behavior at leisure activities where the main reason for attendance is the music. While drawing conclusions is challenging due to the methodological limitations of the included studies, we do find that a mixed attitude towards loud music was identified. Participants seemed to acknowledge the high music volumes being played and actually indicated a preference for lower volumes where the conversation is possible. Nevertheless, participants also declared that they would not avoid a night-club/music venue because of the loud volumes played [2]. These responses highlight the need for the venues to be mindful of their customers' health. A different population studied by Cameron et al. [32] expressed their wish for louder music to be played at the concert they were attending, however, the volumes played during their experiment generally fluctuated between 60 and 80dB, which is considerably lower than the normal volumes played at nightclubs or music venues. The discrepancy in the results found on attitude to loud music can also be explained by a confounder that was not considered in the results: hearing health antecedents. Beach et al. [2] identified important changes when evaluating the impact of tinnitus antecedents and self-perceived risk of the noise levels on their results. Participants who often experience tinnitus or had a high self-perceived risk were significantly more likely to prefer lower music volumes. These two variables can be traced back to the level of education on the risks of loud music, as people suffering from hearing disorders are more likely to educate themselves on the matter [61]. Even though teenagers are particularly at risk of developing NIHL, they are less mindful of the dangers of loud music [3,61]. For instance, Degeest et al [29] demonstrated that high schoolers scored significantly higher on the BAHPHL questionnaire compared to young adults [33]. Indicating an attitude where one does not care about the possible consequences of hearing loss and is unaware of the benefits of HPDs. These results highlight the need to strengthen current education and prevention programs to target youth, particularly in schools or universities.

With this review we identified VLF, high groove, high tempo, and live performances as variables that positively affect participants' recorded movements. More movements in response to the music can be linked to a greater appreciation and increased dancing. Although not directly studied in present study, dancing is a major factor involved in some of the investigated leisure activities such as nightclubs. For instance, dancing can stimulate the production of endorphins, elevating one's moods [62]. Emotions is another outcome that was shown to be heightened by certain musical characteristics. Live performance, high tempo, high groove, songs with happy connotations, and unexpectedness increase the emotional intensity or the participants' sense of happiness. Although Cameron et al. [32] did record high self-rated enjoyment scores at their VLF concert, this outcome was not compared to non-VLF concert scores, thereby limiting the reliability of these

results. The lack of statistical difference between the in-concert VLF on versus off ratings further suggests that VLF does not significantly increase enjoyment. The positive effects of live performances on both movements and emotions can be exemplified by the tendency of crowds to place themselves in front of the DJ booth.

As previously mentioned, nightclub owners are often reluctant to lower music levels due to the hypothesized impact it may have on alcohol consumption. Only two of our included studies investigated the impact of sound levels on alcohol consumption. Since these studies were conducted by the same research team and exhibited some concerns regarding bias, our findings on this topic are limited. Therefore, more research is needed to gain a more definitive understanding of this matter. When evaluating harmful behavior, the review primarily included studies that investigated the effects of music genre. According to our results, specific music genres like classical, hardcore, funk, electronic, pop dance, and forro/zouk were associated with an increase in aggression, sexual assault incidence, and substance consumption. Overall, these results denote the importance that musical characteristics other than volumes can have on one's experience and behavior, which is useful for venue owners who are trying to reduce the sound levels played without decreasing their customer's musical experience. Although our review offered some valuable conclusions, they are not sufficient to provide exact recommendations to venue owners, which highlights the need for more research on how musical characteristics other than sound levels could impact customers—particularly in the context of potential interventions.

The research questions of this review was intentionally broad, as it aimed to explore several relationships between musical characteristics and participants' behaviors and experiences. This breadth allowed us to identify several distinct patterns, detailing outcomes in the following domains: attitude lo loud music, effect on movement, emotions, harmful behaviour and hearing outcomes. However, it also required discussing studies with highly heterogeneous methodologies, measures, and reported outcomes, which limited our ability to pool data or draw overarching conclusions. Nevertheless, this diversity enabled us to formulate specific conclusions from the literature within several individual domains. In some cases, the variability across findings within a single domain further complicated cross-study comparisons. Nonetheless, our comprehensive literature search ensured that a wide range of relevant perspectives were captured, contributing to a richer, more nuanced understanding of the relation between musical characteristics and participants' behaviors and experiences.

Drawing strong conclusions based on this review was also limited by the high risk of bias present in some of the included studies. However, as previously mentioned, the breadth of our review allowed for domain-specific conclusions to be drawn. Because findings were interpreted within their respective domains, the presence of high risk of bias in certain studies did not necessarily affect the validity of the conclusions drawn from other domains. Examples of high risk of bias studies include Theorell et al. [31] who used differing and inconsistently described sample populations across conditions, raising concerns about comparability. Furthermore, several studies used observations as a method to measure their outcome. For example, Forsyth [40] relied on data from two observers, which, given the large sample sizes, may have missed subtle or verbal cues. Carter & Black [35] and Hunter et al. [34] also presented a higher risk of bias, however their outcome was the participants' attitude to loud music which was the most frequently investigated outcome across the included studies, mitigating the impact of bias in these two cases. Nevertheless, the presence of these methodological limitations calls for a cautious interpretation of our results and highlights the need for more rigorous research in this domain, including non-observational standardized outcome measures, clearer reporting of sample characteristics, and more robust methods to minimize bias.

Apart from the risk of bias introduced by the study designs of the majority of studies included in this review, the absence of a standarzided and validated tool for assessing musical experiences limits the quality of the presented outcomes. Besides this, unaddressed confounding factors such as age of the studied population, hearing health history, and frequency of attendance should be taken into account in studies before drawing conclusions on the relationship between the actual music characteristics and the participants' experience and behavior during leisure activities. These factors, while included in the risk of bias assessment, were inconsistently reported across studies and introduced significant heterogeneity. They were not extracted or used in this review based on the exploratory basis of our study, which limits the strength of our conclusions.

With this systematic review, we highlighted the variety of effects that different musical characteristics can have on one's experience and behavior. Although highly amplified music is an important part of the studied leisure activities, participants also acknowledge finding it too loud on certain occasions. This observation, in addition to the considerable risks involved with loud sounds, emphasizes the need to divert the focus to other musical characteristics when wanting to maximize attendees' experiences. Our findings provide new insights into the impact of music on the experience of leisure activity attendees, but more importantly it highlights the lack of adequate studies assessing this topic. In order to reach adequate prevention of hearing damage, and to limit the growing number of individuals with tinnitus and hearing loss we need well performed studies of high quality. These findings can serve as valuable input for shaping future prevention policies. We hope this systematic review will be the starting point for new research.

## Supporting information

**S1 File. Search strings.** Includes search strings for the following databases: PubMed, Cochrane, Embase Elsevier, and PsychInfo.
(DOCX)

**S2 File. Risk of bias assessment criteria.**
(DOCX)

**S3 File. YANS questionnaire.**
(DOCX)

**S1 Checklist. PRISMA checklist.**
(DOCX)

## Author contributions

**Conceptualization:** Adriana Leni Smit, Inge Stegeman.

**Data curation:** Céline Daelemans.

**Formal analysis:** Céline Daelemans.

**Funding acquisition:** Adriana Leni Smit, Inge Stegeman.

**Investigation:** Céline Daelemans, Casper Bonapart.

**Methodology:** Céline Daelemans, Adriana Leni Smit, Inge Stegeman.

**Project administration:** Céline Daelemans.

**Supervision:** Adriana Leni Smit, Inge Stegeman.

**Validation:** Céline Daelemans, Casper Bonapart, Adriana Leni Smit, Inge Stegeman.

**Visualization:** Céline Daelemans.

**Writing – original draft:** Céline Daelemans.

**Writing – review & editing:** Céline Daelemans, Casper Bonapart, Adriana Leni Smit, Inge Stegeman.

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
