## [Decision Letter · Decision Letter 0]

PONE-D-24-55630Temporary fun, permanent damage: a systematic review on the effects of musical characteristics on participants’ experience and behavior during leisure activitiesPLOS ONE

Dear Dr. Daelemans,

Thank you for submitting your manuscript to PLOS ONE. After careful consideration, we feel that it has merit but does not fully meet PLOS ONE’s publication criteria as it currently stands. Therefore, we invite you to submit a revised version of the manuscript that addresses the points raised during the review process.

**Two experts in the field have carefully reviewed the manuscript entitled "Temporary fun, permanent damage: a systematic review on the effects of musical characteristics on participants’ experience and behavior during leisure activities". You can find their comments below. While they both had positive comments on the manuscript, they also had suggestions for improvement and requested clarifications and reworking of some parts. **

**In light of these reviews, I am requesting a major revision and resubmission, in which you will need to respond to each point in each review. **

We look forward to receiving your revised manuscript.

Kind regards,

Bruno Alejandro Mesz, Ph.D.

Academic Editor

PLOS ONE

**Journal Requirements:**

1. When submitting your revision, we need you to address these additional requirements. Please ensure that your manuscript meets PLOS ONE's style requirements, including those for file naming. The PLOS ONE style templates can be found at https://journals.plos.org/plosone/s/file?id=wjVg/PLOSOne_formatting_sample_main_body.pdf and https://journals.plos.org/plosone/s/file?id=ba62/PLOSOne_formatting_sample_title_authors_affiliations.pdf 2. As required by our policy on Data Availability, please ensure your manuscript or supplementary information includes the following:  A numbered table of all studies identified in the literature search, including those that were excluded from the analyses.   For every excluded study, the table should list the reason(s) for exclusion.   If any of the included studies are unpublished, include a link (URL) to the primary source or detailed information about how the content can be accessed.  A table of all data extracted from the primary research sources for the systematic review and/or meta-analysis. The table must include the following information for each study:  Name of data extractors and date of data extraction  Confirmation that the study was eligible to be included in the review.   All data extracted from each study for the reported systematic review and/or meta-analysis that would be needed to replicate your analyses.  If data or supporting information were obtained from another source (e.g. correspondence with the author of the original research article), please provide the source of data and dates on which the data/information were obtained by your research group.  If applicable for your analysis, a table showing the completed risk of bias and quality/certainty assessments for each study or outcome.  Please ensure this is provided for each domain or parameter assessed. For example, if you used the Cochrane risk-of-bias tool for randomized trials, provide answers to each of the signalling questions for each study. If you used GRADE to assess certainty of evidence, provide judgements about each of the quality of evidence factor. This should be provided for each outcome.   An explanation of how missing data were handled.  This information can be included in the main text, supplementary information, or relevant data repository. Please note that providing these underlying data is a requirement for publication in this journal, and if these data are not provided your manuscript might be rejected.   

Reviewers' comments:

Reviewer's Responses to Questions

**Comments to the Author**

1. Is the manuscript technically sound, and do the data support the conclusions?

Reviewer #1: Yes

Reviewer #2: Partly

2. Has the statistical analysis been performed appropriately and rigorously? 

Reviewer #1: N/A

Reviewer #2: Yes

3. Have the authors made all data underlying the findings in their manuscript fully available?

Reviewer #1: Yes

Reviewer #2: Yes

4. Is the manuscript presented in an intelligible fashion and written in standard English?

Reviewer #1: Yes

Reviewer #2: Yes

5. Review Comments to the Author

**Reviewer #1: ** The manuscript is a systematic review on how different musical characteristics influence the experiences and behaviours of individuals engaged in leisure activities. It is well-written and somewhat relevant to the field. The main issue is the inclusion of lab-based studies, which, in my opinion, conflict with the aim of the review. The following specific comments require attention.

Major comments:

• Line 126: does the inclusion of lab-based studies clash with your introduction, results and discussion?

• Line 137: did you conduct the search only on that day? Was it repeated at other times? Who performed the search?

• Pag 20 and 36 are difficult to read as tables overlap the text. Please fix it. Please place the tables in a separate document.

• Line 588-561: I appreciate your protocol but these factors should be discussed.

Minor comments:

• Line 54: has been gaining in popularity -> has gained popularity.

• Line 58: please add a reference for identified as a growing threat.

• Line 64: “permitted” and “up to” are a bit odd. Maybe: enabled these audio devices to reach maximum volumes of 78 to 136 dB.

• Line 72: please add also notch HL.

• Line 81: the sentence “some European country … ones” is too vague.

• Line 84: the campaigns should be in quotes.

• Line 88: maybe add Dolan - Preferred music-listening level in musicians and non-musicians.

• Line 112: thank you for the checklist.

• Line 186: did you encounter any discrepancies in assessing the risk of bias? If so, how were they resolved?

• Line 224: remove [21], no need of it.

• Line 225: please mention the papers with 16 and 3256 participants.

• Line 228: “The most common” should be in a new line.

• Line 228-229: remove bracket.

• Line 231: to improve readability, please list the 9 categories as a bulleted or numbered list.

• Line 347: clarify the YANS and YANS-R abbreviations and add references.

• Line 349: since it’s the only significant value, please add which test and its results.

• Line 351-355: if the sentences are citations, they should be in quotes.

• Line 377-379: please rephrase

• Line 385: “Due to the extensive data, the detailed results were not included in Table 8.” should be a footnote.

• Pag 36: it’s not clear what the numeric adverbs refer to.

• Line 418: here and elsewhere p<0.01, being constant how do you report results please.

• Line 429-430: please change how you report CI as double minus is confusing.

• Line 451: either comma or period.

• Line 454: 9.5 out of?

• Line 463: p<0.03 is odd

• Line 465: euro?

• Line 469: maybe a better title would be along the line of hearing and HPD outcomes

• Line 472-475: please rephrase as it’s not clear.

• Line 478: please elaborate “numerous hearing health issues”

• Line 483: first time BAHPHL used in the text please clarify the abbreviation and add references.

• Line 560: “The poor” should be in a new line.

**Reviewer #2: ** General Comment:

The article presents an interesting approach to the impact of various musical characteristics on participants' experience and behavior in recreational activities. The systematic review of previous studies is valuable, providing a comprehensive overview of the effects of music on health and human behavior. However, in my opinion, the manuscript requires substantial revisions before it can be considered for publication. Below are general comments that the authors could use to improve and strengthen their work.

Major Revisions:

Introduction:

The manuscript offers an extensive review of the literature regarding the factors contributing to hearing damage caused by exposure to elevated sound levels. However, the primary aim of the study is to analyze how different musical characteristics—such as tempo, tonality, rhythm, and sound level—affect individuals' behavior and experience in recreational settings.

My main concern is that the thematic transition between hearing damage and the analysis of musical characteristics is not well articulated, resulting in a lack of cohesion in the structure of the introduction. Although a strong context is provided on the negative effects of high noise levels, the manuscript seems to focus on the impact of musical characteristics on listeners, yet this topic is not adequately developed in this section.

The excessive emphasis on hearing damage may divert the reader’s attention from the true purpose of the study. Furthermore, the introduction lacks a precise explanation of which musical characteristics will be analyzed in detail and a clear justification for their selection.

I recommend that the authors expand the introduction, offering a more detailed argument about the musical characteristics to be discussed, their interaction with the sound environment, and their impact on the recreational experience. This would provide a stronger conceptual framework aligned with the study's objectives, aiding readers’ understanding from the outset.

Methods:

In the data collection section (lines 150-151), the authors state that "Only interventions and outcomes which were relevant to the research question were extracted." However, the exact research question is unclear, making it difficult to understand the strategy used to select the articles. It is crucial that the research question(s) be explicitly stated at the end of the introduction, along with the study's hypotheses, if applicable. This would provide greater clarity regarding the study’s direction and would more precisely justify the selection of the articles included in the review.

Discussion:

In line 551, the authors mention, "The broad research questions of this review allowed for several relationships between musical characteristics and participants’ experiences and behavior to be drawn. However, it also allowed for studies with very heterogeneous data to be included, making it difficult to attain clear conclusions" While it is understandable that the diversity of studies reviewed may complicate deriving solid conclusions, it is essential for the authors to further elaborate on how this heterogeneity might have impacted the interpretation of the findings.

In particular, the lack of validity in many of the reviewed studies (e.g., design issues, biases, and the absence of comparable data) raises serious doubts about the reliability of the conclusions. If the foundation on which this review is built is methodologically flawed, it becomes challenging to assess the robustness of the claims presented.

Additionally, in line 560, the authors acknowledge that "The poor study design of some studies further limited the conclusions", but they do not sufficiently develop how this fact compromises the credibility of the results obtained. It is imperative to discuss in greater depth the relationship between these methodological limitations and the extracted results, as these issues could significantly impact the review's ability to generate reliable and useful conclusions. The reader should be aware of how these shortcomings condition the interpretation of the findings.

Furthermore, in line 545, the authors suggest that "According to our results, specific music genres were associated with an increase in aggression, sexual assault incidence, and substance consumption". While this is an intriguing point, there is a lack of clarity regarding the stance venue owners should take in response to these findings. Although the authors acknowledge that musical characteristics, besides volume, influence attendees' experience and behavior, they do not offer clear recommendations for venue owners. It would be highly beneficial if the authors included more specific suggestions, such as which musical genres could minimize aggressive behavior or substance abuse without negatively impacting the musical experience for clients. These recommendations would add practical and tangible value to the study's findings.

Minor Revisions:

Title:

The current title, "Temporary fun, permanent damage," suggests a direct relationship between fun and permanent damage, which is neither demonstrated nor thoroughly discussed in the review. This statement could lead to misinterpretations of the study’s results and conclusions. A more appropriate title would focus on the topics that are actually addressed, namely, the effects of musical characteristics on participants' experience and behavior during recreational activities. I recommend adjusting the title to more accurately reflect the study's focus and scope, avoiding claims not supported by the evidence presented.

Page 5, last paragraph:

The study’s objective is mentioned twice in the last paragraph of the introduction.

Page 6, line 105:

The authors state, "Understanding the influence that musical characteristics can have on one’s experience at events can aid policymakers in navigating around the public’s opinion to create guidelines for a safer yet equally entertaining experience". In this case, I believe the phrase requires more clarity regarding how exactly this balance between auditory safety and pleasurable experience for attendees could be achieved. The concept of "balancing public opinion" is vague, and it is unclear how musical characteristics relate to the formation of public policy in this context. It would be useful to specify how data on musical characteristics could be concretely used to influence policymakers' decisions, especially regarding setting volume limits or regulations on the type of music and its impact on auditory health. This would help connect the research more directly to practical and policy applications.

Methods:

Page 7, line 150:

The authors mention: "If these effect measures were not reported in the article according to the interventions of interest, the corresponding authors were contacted to obtain the raw data. " It would be advisable to specify how many cases involved contacting authors and how many of these resulted in effective responses.

Page 9, line 190:

In the section on effect measures, the authors state: "If these effect measures were not reported in the article according to the interventions of interest, corresponding authors were contacted for raw data." It would be helpful to specify how many cases involved contacting authors for raw data and how many attempts resulted in obtaining the required information.

Page 10, line 210:

The authors state: "After the full texts of the articles were retrieved and assessed for eligibility, 44 were excluded mainly due to wrong outcomes or designs, and others for alternative reasons (see Figure 1)" It would be helpful for the authors to specify the criteria used to determine that these articles had "incorrect designs."

Results:

It would be advisable for the authors to consider condensing some of the results presented in tables into figures, as this would facilitate a quicker and clearer understanding of the findings for readers. Figures, such as graphs or diagrams, can highlight patterns, trends, and comparisons more visually and accessibly than tables. Additionally, figures allow relationships between different variables to be captured at a glance, expediting the interpretation process without sacrificing data precision.

6. PLOS authors have the option to publish the peer review history of their article (what does this mean? ). If published, this will include your full peer review and any attached files.

**Do you want your identity to be public for this peer review?** For information about this choice, including consent withdrawal, please see our Privacy Policy .

Reviewer #1: No

Reviewer #2: No

---

## [Author Response · Author response to Decision Letter 1]

18 May 2025

Dear editor and reviewers,

We would like to thank the reviewers for their comments on our manuscript entitled “It’s all in the music: a systematic review on the effects of musical characteristics on participants’ experience and behavior during leisure activities”. We have revised our manuscript, data, and supporting information in line with your suggestions.

Our detailed responses to the reviewers’ comments can be found in the attached word document titled "Response to Reviewers".

Thank you for considering our manuscript for publication.

Sincerely,

On behalf of the author-team,

Céline Daelemans

---

## [Decision Letter · Decision Letter 1]

PONE-D-24-55630R1**It's all in the music:**  a systematic review on the effects of musical characteristics on participants' experience and behavior during leisure activitiesPLOS ONE

Dear Dr. Daelemans,

Thank you for submitting your manuscript to PLOS ONE. After careful consideration, we feel that it has merit but does not fully meet PLOS ONE’s publication criteria as it currently stands. Therefore, we invite you to submit a revised version of the manuscript that addresses the points raised during the review process.

Dear authors, tthank you for submitting your revision. I have read your revised manuscript, with the last round of reviewers’

comments in hand. Thank you for your careful attention to these corrections,

additions, and clarifications. I will not send your revision back to them for another reading, but I am requesting you to address the minor points of Reviewer #1.

We look forward to receiving your revised manuscript.

Kind regards,

Bruno Alejandro Mesz, Ph.D.

Academic Editor

PLOS ONE

Journal Requirements:

Reviewers' comments:

Reviewer's Responses to Questions

**Comments to the Author**

1. If the authors have adequately addressed your comments raised in a previous round of review and you feel that this manuscript is now acceptable for publication, you may indicate that here to bypass the “Comments to the Author” section, enter your conflict of interest statement in the “Confidential to Editor” section, and submit your "Accept" recommendation.

Reviewer #1: All comments have been addressed

Reviewer #2: All comments have been addressed

2. Is the manuscript technically sound, and do the data support the conclusions?

Reviewer #1: Yes

Reviewer #2: Yes

3. Has the statistical analysis been performed appropriately and rigorously? 

Reviewer #1: N/A

Reviewer #2: Yes

4. Have the authors made all data underlying the findings in their manuscript fully available?

Reviewer #1: Yes

Reviewer #2: Yes

5. Is the manuscript presented in an intelligible fashion and written in standard English?

Reviewer #1: Yes

Reviewer #2: Yes

6. Review Comments to the Author

Reviewer #1: The authors did an extensive revision on the manuscript and addressed many of the raised points by the reviewers. So, the manuscript improved substantially.

Points that need to be addressed:

Lines 92-93: “several studies” but only one was mentioned. Furthermore, several of the statements of the introduction are based on [2] Beach and Gilliver “Time to listen”. Please find other sources to corroborate some of them.

Line 115: remove “low”

Line 119: in participants of leisure → among participants; affect → influence

Lines 186-199: “The extraction form was composed of eight sections.” but only four sections are clearly marked.

Line 244: “wrong outcomes”. It’s wrong for your aim, not for their studies. Please change the adjective.

Fig 1: as above, change “wrong”.

Line 283 and Table 1: “We did not specifically note in Table 1 that these studies were conducted in lab settings”. I disagree. I think this information is needed for comprehensiveness.

Lines 288-289: please add reasoning why you only included low groove/low tempo. In the results (line 429) and Table 8 you mentioned also high groove and tempo.

Table 1: several studies did not report age. How confident are you that the study did not include participants <15 years old?

Lines 300-301, 303-304, 311-312, and 316-317: remove new lines.

Tables 2 and 3: Remove the column “quality” as it was not in the NOS. The list as table note isn’t easy to digest. Similar for the other tables.

Line 393-395 : no need of F1, F2 etc. later factor 1 → first factor.

Line 403: “Black”. Remove full stop.

Table 7 and others: For the notes, please use the conventional symbol order so readers don’t need to count the number of asterisks. See https://en.wikipedia.org/wiki/Note_(typography)#Numbering_and_symbols

Table 8: what VLF, PC1, and PC2 stand for?

Lines 453 and 455: I appreciate the precision in reporting the p-values but using “p>0.05” would improve the flow.

Line 484: which other two articles?

Reviewer #2: Dear Bruno,

I have reviewed the authors’ responses to my comments and the revised manuscript. All my concerns have been satisfactorily addressed, and the manuscript has been improved accordingly. The study now meets the journal’s standards for clarity, rigor, and presentation.

I recommend acceptance in its current form.

Best regards,

7. PLOS authors have the option to publish the peer review history of their article (what does this mean? ). If published, this will include your full peer review and any attached files.

**Do you want your identity to be public for this peer review?** For information about this choice, including consent withdrawal, please see our Privacy Policy .

Reviewer #1: No

Reviewer #2: No

---

## [Author Response · Author response to Decision Letter 2]

30 Jun 2025

Dear editor,

We would like to thank the reviewers for their comments on our manuscript entitled “It’s all in the music: a systematic review on the effects of musical characteristics on participants’ experience and behavior during leisure activities”.

Our detailed responses to the editor and reviewers comments can be found in the document labeled 'Response to Reviewers'.

Thank you for considering our manuscript for publication.

Sincerely,

On behalf of the author-team,

Céline Daelemans

---

## [Editor Report · Decision Letter 2]

**It's all in the music:**  a systematic review on the effects of musical characteristics on participants' experience and behavior during leisure activities

PONE-D-24-55630R2

Dear Dr. Daelemans,

We’re pleased to inform you that your manuscript has been judged scientifically suitable for publication and will be formally accepted for publication once it meets all outstanding technical requirements.

Kind regards,

Bruno Alejandro Mesz, Ph.D.

Academic Editor

PLOS ONE
---

## [Editor Report · Acceptance letter]

PONE-D-24-55630R2

PLOS ONE

Dear Dr. Daelemans,

I'm pleased to inform you that your manuscript has been deemed suitable for publication in PLOS ONE. Congratulations! Your manuscript is now being handed over to our production team.

Kind regards,

on behalf of

Dr. Bruno Alejandro Mesz

Academic Editor

PLOS ONE